# Leveraging Low Rank Structure in the Lazy Regime

## Abstract

Understanding the training dynamics of neural networks has gained much interest in the scientific community. The dynamics of training over-parameterized models is characterized by the lazy regime in which networks exhibit near-linear behavior and minimal parameter changes. In addition, it has been argued that the Jacobian of large neural models has a low-rank structure. In this paper, we focus on the opportunities laid out by the combination of low-rankness and laziness of large neural models. Specifically, we provide a scalable way to measure the extent of laziness, evaluated via the rate of change of the model Jacobian, as well as a scalable method to verify low-rankness of the model Jacobian without storing the entire Jacobian. Taking advantages of both laziness and low-rankness, we design a scalable training algorithm for over-parameterized models that performs backpropagation-free gradient descend training. In particular, this algorithm is of lower computation and storage requirements in cases of massive parameter sharing, as is the case of many state-of-the-art neural architectures. Empirical results confirm the scalability and effectiveness of our approach, opening new pathways for exploring novel learning strategies in neural networks.

## 1 Introduction

Understanding the training dynamics of neural networks is essential for uncovering how they operate and learn. The empirical success of neural networks has far outpaced the theoretical understanding of their underlying mechanisms, yet ongoing research aims to identify the factors that enable effective learning. Recent studies have identified two distinct training regimes: *lazy* and *active*, as well as mixtures of the two. Each regime offers unique insights into how networks learn and adapt, with factors such as network width, initialization, and training duration playing key roles in shaping a model's behavior (Chizat et al., 2019; Lee et al., 2019; Tu et al., 2024).

The lazy regime, in particular, has gained attention due to its simplified, linear dynamics, where networks rely on a fixed, nearly constant kernel during training. This contrasts with the active regime, where the network significantly updates its internal representations and adapts features over time. Understanding when a network operates in the lazy regime offers valuable insights into how models can achieve learning with minimal parameter updates, while the active regime involves more flexible, feature-adapting behavior that may enhance learning capacity.

Although the lazy regime is theoretically appealing due to its linear dynamics and minimal parameter changes, its practical application is constrained by the requirement for extremely wide networks. These large networks widths result in computational and storage challenges, particularly due to the Jacobian matrix, whose dimensions scale with both the number of samples $n$ and the network width $m$. While the near-constancy of the Jacobian suggests that storing it could eliminate the need for recomputation during each step of backpropagation, this approach becomes impractical in large-scale settings without leveraging the matrix's low-rank structure. The high dimensionality of the Jacobian makes direct storage infeasible unless we can effectively compress it, which would require exploiting inherent low-rank properties. If we can leverage the low-rank nature of the Jacobian in wide networks, it would enable scalable algorithms that preserve the benefits of linear dynamics while significantly reducing computational overhead. To the best of our knowledge, the potential for integrating low-rank structures within the lazy regime has not yet been explored. Such a combination could unlock of more efficient training methods.

To address this challenge, we propose a novel framework that connects the low-rank structure of the Jacobian with the lazy regime of neural networks. Our approach tackles the computational inefficiencies of traditional methods by introducing efficient techniques for estimating the rank of the Jacobian using a carefully selected reference set, and validating the laziness of a network by calculating the rate of change of the Jacobian. Instead of storing and processing the full Jacobian matrix, which is often infeasible for large networks, we exploit the low-rank property to significantly reduce dimensionality. Specifically, we select a small reference set $\mathcal{R}$ of important weights and compute only the sub-matrix of the Jacobian corresponding to this subset. By focusing on the most significant eigenvectors of this sub-matrix, we approximate the full Jacobian's rank without the need for costly storage or computation.

Building on these insights, we introduce a backpropagation-free learning algorithm grounded in Neural Tangent Kernel (NTK) theory. This novel algorithm simplifies training by exploiting both the low-rank structure and the lazy regime, enabling robust learning dynamics with reduced computational overhead. Our method is particularly well-suited for large parameter-sharing networks, such as convolutional neural networks (CNNs) and transformers, as it scales efficiently while maintaining performance.

In summary, our contributions are threefold: (1) we introduce a reliable method for estimating the Jacobian's rank, (2) we provide a concrete approach to determining network laziness. (3) These innovations allow us to construct a backpropagation-free learning algorithm that leverages the stability and efficiency of low-rank structures in the lazy regime, opening new pathways for scalable and efficient machine learning models.

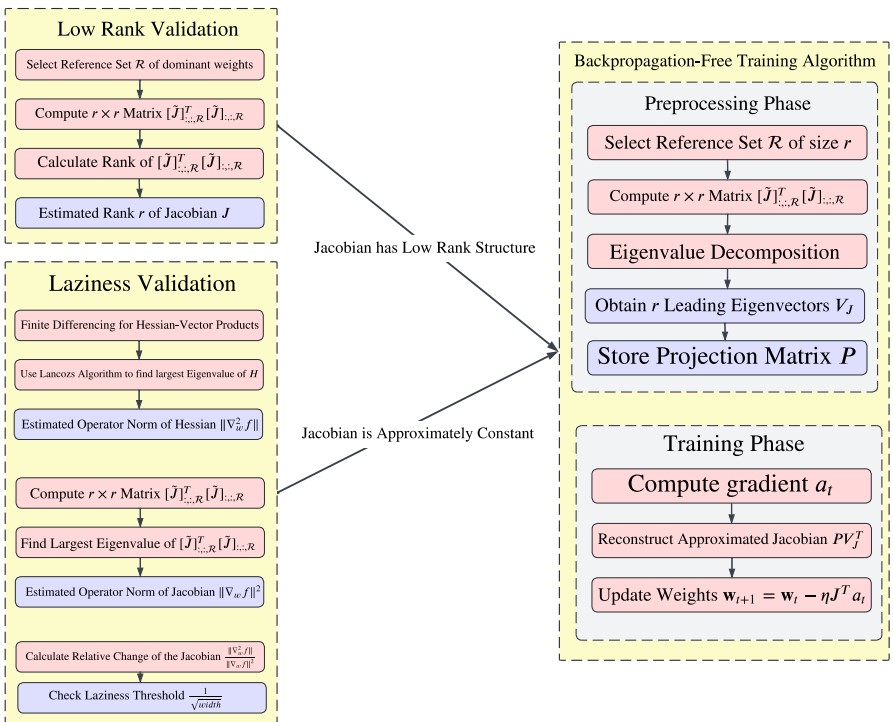

Figure 1: **Overview of the Backpropagation-Free Training Algorithm Using Lazy and Low-Rank Validation.** The process begins by validating whether the network exhibits lazy training behavior and a low-rank Jacobian structure through relative change calculations and eigenvalue analysis. The algorithm leverages a precomputed Jacobian sub-matrix, constructed using a reference set of weights. These computations are done during the preprocessing phase, followed by training without traditional backpropagation. Instead, the weight updates are performed using the approximated low-rank Jacobian, significantly reducing computational complexity while preserving training efficiency in large models.

## 1.1 Notations

Let $D = \left\{ \left( x^{(1)}, y^{(1)} \right), \ldots, \left( x^{(n)}, y^{(n)} \right) \right\}$ be a dataset consisting of $n$ pairs of inputs and outputs, where for all $i$, $x^{(i)} \in \mathbb{R}^{d_x}$ and $y^{(i)} \in \mathbb{R}^{d_y}$. We consider a neural network model $f$ with output $f(w; x) \in \mathbb{R}^{d_y}$, having $m$ total parameters, which we collectively denote as $w$. With a slight abuse of notation, we use subscripts to indicate vector entries or time indices, depending on the context, and superscripts to indicate specific data instances. Let $\ell(y, f(w; x))$ be a loss function associated with an individual input-output pair $(x, y)$ obtained using parameters $w$, and define the loss over a subset $S \subseteq D$ of the entire dataset as the average of the individual losses, i.e.,

$$\ell(w; S) = \frac{1}{|S|} \sum_{\left( x^{(i)}, y^{(i)} \right) \in S} \ell \left( (y^{(i)}, f\left( w; x^{(i)} \right) \right). \tag{1}$$

We denote the model's Jacobian at time $t$ by $J_t$, i.e., $[J_t]_{i,j,k} = \left. \frac{\partial}{\partial w_{t,k}} f_j \right|_{w_t, x^{(i)}}$, where $f_j$ is the $j$-th entry of the model output, $w_t$ are the parameters at time $t$, and $w_{t,k}$ is the $k$-th parameter. In addition, we denote the gradient of the loss with respect to the model output at time $t$ by $a_t$, i.e., $[a_t]_{i,j} = \left. \frac{\partial}{\partial f_j} \ell \right|_{w_t, \left( x^{(i)}, y^{(i)} \right)}$.

The Neural Tangent Kernel (NTK) at time $t = 0$ is denoted:

$$K_{NTK}(i, j) = \langle [J_0]_{i,:,:}, [J_0]_{j,:,:} \rangle, \tag{2}$$

## 2 Previous Work

**Neural Tangent Kernel.**   The remarkable empirical success of deep neural networks has driven extensive research to understand their underlying mechanisms. The neural tangent kernel (NTK), introduced in (Jacot et al., 2018), marks a key milestone, as it facilitates the use of well-developed theoretical tools of kernels. Broadly, the NTK is defined as the inner product of the gradients of a network's output of $f(w; x)$ with respect to its trainable parameters w: $K(x, z)(w) := \nabla_w f(w; x)^T \nabla_w f(w; z)$, for fixed inputs $x, z \in R^{d_x}$.

In the infinite width limit for certain architectures, the NTK remains constant during training, leading to linear dynamics, often referred to as "lazy training" (Chizat et al., 2019; Lee et al., 2019), where minimal parameter changes keep the network near its linearization. Then the NTK at time t = 0 is denoted $K_{NTK}(i, j) = \langle [J_0]_{i,:,:}, [J_0]_{j,:,:} \rangle$, as declared in section 1.1. This constancy of the NTK allows for provable optimization guarantees. Specifically, studies have shown that gradient descent converges to a global minimum at a linear rate in the NTK regime, even for over-parameterized networks (Du et al., 2018; 2019; Arora et al., 2019). More recently, it has been demonstrated that such over-parameterized neural networks achieve optimal classification power (Radhakrishnan et al., 2022) and exhibit robustness to noise (Belkin et al., 2019). Our work aims to develop a practical algorithm derived from NTK theory that maintains the equivalent mathematical guarantees provided by these theoretical analyses.

**Disentangling Different Training Regime.**   Further studies have explored the conditions under which networks exhibit linear or nonlinear dynamics during training, influenced by network width and initialization. Wide networks are found to evolve into two distinct regimes: the "lazy training" regime with near-linear dynamics and minimal NTK change or the "active/feature training" regime where the NTK evolves in time and learns features.

Recent research has refined this dichotomy, emphasizing the interplay of network width, depth, initialization, and the learning task, offering frameworks to analyze complex neural network dynamics across various architectures and conditions. Particularly, (Tu et al., 2024) offers a fine-grained analysis of linear neural networks, showing that different parts of a network can operate in distinct regimes simultaneously, suggesting a "mixed" dynamics regime.

(Geiger et al., 2020) empirically demonstrates the transition between the NTK and mean-field regimes, is governed by a scaling parameter of the last layer that varies with the square root of network width. (Liu et al., 2020) offers a novel viewpoint on the "transition to linearity," associating

it with the scaling of the Hessian norm as network width increases. They challenge the traditional "lazy training" explanation, highlighting that constancy may not hold even when individual parameters change only slightly. Their analysis reveals that constancy of the Neural Tangent Kernel (NTK) is instead tied to structural attributes of the network, such as output layer linearity, and that the Hessian norm scales as $\frac{1}{\sqrt{width}}$. Our empirical results are consistent with their theoretical findings, as we observe similar scaling behavior of the Hessian norm in our experiments. We extend their analysis by introducing a scalable, empirical method utilizing the Lanczos algorithm and finite differencing to compute Hessian-vector products. This allows us to quantitatively measure the degree of laziness across various settings, enhancing our method's applicability to large-scale models.

**Low-Rank Structure in Lazy Regime.**  (Oymak et al., 2019) empirically demonstrates that over-parameterized neural networks exhibit a Jacobian with low-rank properties, characterized by a few large singular values and many smaller ones. This low-rank structure defines a low-dimensional "information space," where learning occurs rapidly since the majority of the label vector resides in this space. Conversely, label noise tends to project onto the "nuisance space," corresponding to smaller singular values, hindering optimization and generalization. While Oymak's method provides a solid framework for understanding the generalization capabilities of over-parameterized networks, its scalability can be limited due to the computational overhead of singular value decomposition (SVD) for large networks. Our method addresses this issue by offering a scalable approach that does not require storing the full Jacobian.

In a similar vein, recent works have leveraged low-rank structures to enhance neural network efficiency and reduce redundancy. For example, LoRA (Low-Rank Adaptation), introduced by (Hu et al., 2021), decomposes weight updates during fine-tuning into low-rank matrices, significantly reducing the number of trainable parameters without sacrificing performance. Inspired by LoRA, (Hao et al., 2024) suggested that gradients can also be compressed into a low-rank subspace, and proposed using random projections to compress the gradients, further improving memory efficiency during training.

Recent studies have demonstrated that learning primarily occurs within a significantly low-dimensional parameter subspace (Gur-Ari et al., 2018; Larsen et al., 2022), promoting a special type of learning known as subspace learning, where model weights are optimized within this low-rank subspace. This finding supports the growing body of work suggesting that gradient matrices are naturally low-rank during training (Zhao et al., 2022; Cosson et al., 2023; Yang et al., 2023). The low-rank property of gradients has been effectively applied to reduce communication costs in distributed training (Wang et al., 2018; Vogels et al., 2020), as well as to lower memory footprints during the training of large models (Gooneratne et al., 2020; Huang et al., 2023; Modoranu et al., 2023).

Building on the ideas of gradient low-rank projection, GaLore (Zhao et al., 2024) introduces a generalized framework for memory-efficient training by dynamically adjusting the rank of gradient representations, reducing memory overhead in optimizing large language models while preserving full-parameter learning dynamics. A key contribution of GaLore is the theoretical proof that the gradient matrix becomes low-rank during training in reversible networks, which supports the method's efficiency. However, GaLore's primary limitation is its reliance on reversible networks, which induce the low-rank gradient structure by design. In contrast, our method does not impose this architectural constraint. Instead, we leverage both the constancy and the low-rank structure of the Jacobian during training in the lazy regime to eliminate redundant representations. This allows us to develop a more versatile and robust framework for low-rank approximation, applicable across a wider range of architectures, optimizing both computational efficiency and stability in backpropagation-free training.

**Backpropagation-free Methods**  Backpropagation is undoubtedly the main approach to train neural networks, yet it is often considered biologically implausible. Consequently, numerous alternatives have been proposed from a biologically inspired perspective (Lillicrap et al., 2016; Liao et al., 2016; Moskovitz et al., 2018; Nøkland, 2016; Kohan et al., 2018; Balduzzi et al., 2015; Choromanska et al., 2019). Other approaches avoid backpropagation by allowing training of all layers in parallel via an information bottleneck criterion (Ma et al., 2020), or by using random weight updates that are approximately orthogonal to the gradient (Baydin et al., 2022; Silver et al., 2021).

Closer to our work is Boopathy & Fiete (2022), which leverages NTK theory, relying on the fact that biological neural networks are wide and shallow, compared to artificial neural networks. They showed that under the NTK regime, there is an alignment between the correlation of the activations and the dynamic of the weight changes, and propose simplified training algorithms utilizing this alignment, that are theoretically equivalent to gradient descent under the NTK regime.

From an algebraic perspective, (Radhakrishnan et al., 2024) introduces the Deep Neural Feature Ansatz, focusing on feature learning mechanisms in deep networks within the active regime. Emphasizing network structure over parameter updates, this backpropagation-free approach enhances interpretability and efficiency, bridging classical kernel methods and modern neural networks. Our work also comes from an algebraic perspective but operates in the lazy regime.

Our work stands out in its novelty by being the first to leverage both NTK theory and the low-rank structure of the Jacobian in large neural networks. This dual focus enables us to provide a mathematically grounded and computationally efficient backpropagation-free training method that operates within the lazy regime. Furthermore, unlike previous works, we extend beyond theoretical contributions by offering practical tools to verify these key assumptions for specific models and problems at any arbitrary model width. This makes our approach highly adaptable to a wide range of real-world applications.

## 3 UNDERSTANDING NEURAL NETWORKS IN THE LAZY REGIME

In this section, we outline the key assumptions underlying our approach to understanding neural networks in the lazy regime. We begin by proposing an algebraic perspective of laziness, focusing on the low-rank structure of the Jacobian matrix. We then provide empirical methods to verify both the laziness and low-rank assumptions, offering practical tools to assess these properties in large-scale models. Finally, leveraging these assumptions, we present a computationally efficient backpropagation-free training algorithm, which enables training neural networks using a reference set, eliminating the need for traditional backpropagation while maintaining performance.

The main perspective of the proposed research begins with the standard weight update rule of neural networks: $w_{t+1} = w_t - \nabla_w \ell(w; S)|_{w_t}$. Applying the chain rule, we have $\nabla_w \ell = \nabla_w f(w; x) \nabla_f \ell$, which leads to

$$w_{t+1} = w_t - \frac{1}{|S|} \sum_{(x^{(i)}, y^{(i)}) \in S} ([J_t]_{i,:,:})^T [a_t]_{i,:}, \tag{3}$$

where $[J_t]_{i,:,:}$ is the model's Jacobian at $x^{(i)}$ of size $d_y \times m$, and $[a_t]_{i,:}$ is the loss gradient with respect to the model output at $(x^{(i)}, y^{(i)})$ of size $d_y \times 1$.

In the lazy regime, as indicated by NTK theory, the model can be approximated as linear in the weights due to minimal changes in the weights during training. Specifically, we consider the first-order Taylor expansion of the model around the initial parameters $w_0$:

$$f(w; x) \approx f(w_0; x) + J(w - w_0), \tag{4}$$

where $J = \nabla_w f(w; x)|_{w_0}$. This linear approximation implies that $[J_t]_{i,:,:} \approx [J_0]_{i,:,:}$ remains nearly constant over time. Therefore, the weight update in Equation equation 4 can be recast as

$$w_{t+1} = w_t - \frac{1}{|S|} \sum_{(x^{(i)}, y^{(i)}) \in S} ([J]_{i,:,:})^T [a_t]_{i,:}, \tag{5}$$

where the dependency of the Jacobian in $t$ is removed. The weight update in equation 5 is presented as a product of a time-varying term, $[a_t]_{i,:}$, which requires differentiation with respect to the output layer only, and a constant term $[J]_{i,:,:}$. This factorization is the cornerstone of our proposed research.

### 3.1 ALGEBRAIC PERSPECTIVE OF LAZINESS: THE ASSUMPTION OF LOW-RANK JACOBIAN

The assumption that, in large models, the model Jabocian $J$ has a low-rank structure Oymak et al. (2019) gives rise to the following idea. For each output unit $j$, we view $[J]_{:,j,k}$ as an $n$-dimensional representation of the $k$'th weight, obtained by the gradient of the model output w.r.t this weight for

every train instance. We postulate that this is a redundant representation, as we expect that gradients in neighboring data points will not differ significantly from each other. To exploit this assumption, we propose to reduce the dimension from $n$ to some low, fixed dimension $r \ll n$ using the principal directions, i.e., the leading eigenvectors, of the $n \times n$ covariance matrix of the model weights, $[\tilde{J}]_{:,j,:}[\tilde{J}]_{:,j,:}^T$, where $\tilde{\cdot}$ corresponds to row-centering. Computing these principal directions directly is infeasible because $n$ is large, and therefore, the covariance matrix is too large to store and decompose. Instead, we propose a remedy that takes advantage of the low-rank assumption. Specifically, the low-rank assumption implies that the model behavior is governed by a small number of weights (e.g., see Frankle & Carbin (2018)). Therefore, we argue that the covariance can be approximated using a small reference set $\mathcal{R}$ of a fixed size $r$ of important weights, i.e., $[\tilde{J}]_{:,j,:}[\tilde{J}]_{:,j,:}^T \approx [\tilde{J}]_{:,j,\mathcal{R}}[\tilde{J}]_{:,j,\mathcal{R}}^T$ and consider only the sub-matrix $[J]_{:,j,\mathcal{R}}$, of size $n \times r$. Next, we utilize the following elementary property from linear algebra.

**Proposition 3.1.** *Let $A \in \mathbb{R}^{n \times r}$ be a matrix and $n > r$. If $\phi \in \mathbb{R}^n$ is an eigenvector of the matrix $AA^T \in \mathbb{R}^{n \times n}$ with an eigenvalue $\lambda \neq 0$, then $\mathbb{R}^r \ni \psi := \frac{A^T \phi}{\sqrt{\lambda}}$ is an eigenvector of $A^T A \in \mathbb{R}^{r \times r}$ with the same eigenvalue $\lambda$.*

This property implies that the eigenvectors of the $n \times n$ covariance matrix $[\tilde{J}]_{:,j,\mathcal{R}}[\tilde{J}]_{:,j,\mathcal{R}}^T$ can be found using eigendecomposition of the $r \times r$ matrix $[\tilde{J}]_{:,j,\mathcal{R}}^T[\tilde{J}]_{:,j,\mathcal{R}}$, which is much smaller subject to selecting a small reference set, i.e., $r \ll n$. This way we store the $r$ leading eigenvectors of dimension $n$ and the $r$-dimensional reduced representation of the $m$ weights. This idea will be further developed into an efficient training algorithm in Section 3.3.

## 3.2 EMPIRICAL VERIFICATION OF THE LAZINESS AND LOW-RANK ASSUMPTIONS.

The proposed framework relies on two mathematical assumptions regarding the Jacobian: (i) "laziness": the Jacobian remains nearly constant throughout training, and (ii) the Jacobian is low-rank. We now present methods to quantitatively measure the extent to which these assumptions are satisfied in a given setting consisting of a model and training data.

**Laziness.** Following Chizat et al. (2019), we quantify the "laziness" of a neural network by measuring the rate of change of the Jacobian matrix. Specifically, we calculate the ratio $\frac{\|\nabla_w^2 f\|}{\|\nabla_w f\|^2}$, where $f = f_j(:,x)$ represents the output corresponding to a single output unit $j$ and data point $x$ at a time, and the norms are operator norms.

To compute the numerator $\|\nabla_w^2 f\|$, we utilize the Lanczos algorithm Lanczos (1950), an improved version of the power iteration method. The Lanczos algorithm is particularly advantageous because it efficiently finds the eigenvalue of the Hessian with the largest magnitude without requiring the explicit storage of the Hessian matrix itself. This is crucial for large neural networks where the Hessian is too large to store in memory. Instead, the algorithm rather only performs the Hessian-vector products, which we approximate using finite differences as follows:

$$\nabla_w^2 f v \approx \frac{\nabla_w f(w + \epsilon v) - \nabla_w f(w - \epsilon v)}{2\epsilon},$$

where $v$ is a vector, and $\epsilon$ is a small scalar for finite differencing. This approximation introduces an error of $O(\epsilon^2)$ which is generally acceptable for most practical applications. By employing this method, we can efficiently estimate the largest eigenvalue of the Hessian operator without needing to explicitly compute the full Hessian, making it scalable even for very large models. We compute this value for multiple data points $x$ and output units $j$ to obtain an average estimate of the Hessian norm across the network.

For the denominator, we leverage a result from Proposition 3.1 which implies that we can compute the square root of the largest eigenvalue of the $r \times r$ matrix $[\tilde{J}]_{:,j,\mathcal{R}}^T[\tilde{J}]_{:,j,\mathcal{R}}$, where $[\tilde{J}]$ represents the Jacobian of the network. This computation is relatively straightforward and does not pose scalability issues. The results obtained using this method are shown in figure 3.

**Low-rank Jacobian.** To verify the low-rank assumption from (Oymak et al., 2019), we present the effective numerical rank of the $r \times r$ covariance matrix $[\tilde{J}]_{:,j,\mathcal{R}}^T[\tilde{J}]_{:,j,\mathcal{R}}$, as a function of the reference

set size $r$. According to proposition 3.1 this is also equivalent to the rank of $[\tilde{J}]_{:,j,\mathcal{R}}[\tilde{J}]_{:,j,\mathcal{R}}^T$. The effective numerical rank is determined by counting the eigenvalues of $[\tilde{J}]_{:,j,\mathcal{R}}^T[\tilde{J}]_{:,j,\mathcal{R}}$ that are greater than the threshold $r\epsilon : \sigma_1$, where r is the matrix dimension, $\sigma_1$ is its largest eigenvalue and epsilon the machine precision.

This approach allows us to avoid the impractical computations of the full rank of the $m \times n$ Jacobian. Instead, we leverage the fact that the largest $r$ singular values correspond to the square root of the eigenvalues of $[\tilde{J}]_{:,j,\mathcal{R}}^T[\tilde{J}]_{:,j,\mathcal{R}}$.

By selecting a sufficiently large reference set of size $R$, greater than the allegedly Jacobian's rank, the rank of $[\tilde{J}]_{:,j,\mathcal{R}}^T[\tilde{J}]_{:,j,\mathcal{R}}$. will reflect the true rank of the Jacobian. Specifically, $[\tilde{J}]_{:,j,\mathcal{R}}^T[\tilde{J}]_{:,j,\mathcal{R}}$. will have $rank(J)$ non-zero eigenvalues out of R dimention.

When the Jacobian has a low-rank structure, the *rank-to-model size* ratio tends to approach zero as the model size increases because the rank grows much more slowly than the number of parameters. Conversely, for models without a low-rank Jacobian, this ratio remains relatively high, indicating that a significant portion of the model's parameters contribute to the rank. The results supporting this method are shown in figure 2.

### 3.3 EFFICIENT BACKPROPAGATION FREE ALGORITHM LEVERAGES THE ASSUMPTIONS

Our goal is to develop an efficient and theoretically sound backpropagation-free training algorithm that exploits the laziness and low-rank assumptions of the Jacobian matrix.

**Rationale of the Algorithm**   When the laziness assumption holds, the model's Jacobian $J$ remains approximately constant during training. This allows us to precompute $J$ once at initialization and reuse it throughout the training process. However, for large-scale models— which is typically the case since NTK theory requires wide networks— directly storing the full Jacobian of size $n \times m$, where $m$ is the number of parameters and $n$ is the number of data points, is impractical due to the $O(nm)$ storage requirement.

To address this limitation, we exploit the low-rank structure of the Jacobian. Specifically, we approximate the Jacobian using a small reference set of weights and leverage principal component analysis (PCA) to reduce its dimensionality. Importantly, this is done without storing the full Jacobian, making our method scalable to models of any arbitrary width.

**Backpropagation-Free Training Algorithm**   The algorithm comprises two phases as follows:

---
**Algorithm 1** Train without backprop using a reference set

---
**Require:** An initialized neural network model $f$, a loss function $\ell$
  **Pre-processing phase:**
  **for** each output unit $j$ **do**
      Select a small reference set $\mathcal{R}$ of weights of size $r$.
      Compute $[J]_{:,j,\mathcal{R}}$, as a $n \times r$ matrix, selecting the columns of $[J]_{:,j,:}$ corresponding to $\mathcal{R}$.
      Center the rows of $[J]_{:,j,\mathcal{R}}$.
      Obtain the leading $r$ eigenvectors $V_J$ of $[\tilde{J}]_{:,j,\mathcal{R}}[\tilde{J}]_{:,j,\mathcal{R}}^T$ via eigendecomposition of $[\tilde{J}]_{:,j,\mathcal{R}}^T[\tilde{J}]_{:,j,\mathcal{R}}$
      Store $V_J$, and the projection $P := [J]_{:,j,\mathcal{R}}^T V_J$ of each weight onto $V_J$.
  **end for**

  **No backprop phase:**
  **for** each training batch $S$ **do**
      Compute $[a_t]_{i,:}$ for each $i$ s.t. $\left(x^{(i)}, y^{(i)}\right) \in S$.
      Reconstruct the Jacobian $[J]_{S,j,:} := PV_J^T$ for $S$ for each unit $j$
      Update the network weight vector via equation equation 5.
  **end for**
  **return** trained model

---

**Justification of the Approximation**    The approximation relies on the low-rank structure of the Jacobian, which we have empirically validated. By capturing the most significant directions of variation in the Jacobian through PCA, we ensure that the essential information for training is preserved.

Specifically, the true gradient of the loss with respect to the weights at time $t$ , denoted as $g_{\text{true}}$ is given by:

$$g_{\text{true}} := J a_t$$

Let $\tilde{J} = U\Sigma V^T$ be the singular value decomposition of the centered constant Jacobaian $\tilde{J}$. Since V is a unitary matrix, $g_{\text{true}}$ can also be written as

$$g_{\text{true}} = \tilde{J}VV^T a_t$$

Now, leveraging the low rank assumption we validated, instead of using the eigenvectors $V$ of $\tilde{J}\tilde{J}^T$, we use the eigenvectors $V_J$ of $[\tilde{J}]_{:,j,\mathcal{R}}[\tilde{J}]_{:,j,\mathcal{R}}^T$. The gradient is then approximated using

$$g_{\text{approx}} := \tilde{J}V_J V_J^T a_t$$

The approximation is thus accurate whenever the projections of $a_t$ onto $V_J$ and onto $V$ are similar. We can think of this as a partition of the columns space of $U$ to two subspaces: the column space of $U_J$ (the "information subspace") and the orthogonal complement subspace (the "noise subspace").

**Complexity analysis.**    The storage and time complexity of Algorithm 1 are as follows. For each output unit $j$, the eigenvectors matrix $V_J$ is of size $r \times r$ and the projection matrix $P$ is of size $m \times r$. Both can be computed by looping over the dataset of size $n$, one data point at a time. Hence, Algorithm 1 requires $O(m+n)$ storage, considering $d_y$ and $r$ as constants. For a constant minibatch size, reconstructing the Jacobian and computing the weight update take $O(m)$ time, which is similar to the complexity of standard backpropagation over fully connected layers.

**Reduction of computation burden in LLMs, CNNs and RNNs**    While the running time analysis of Algorithm 1 indicates a time complexity comparable to that of training multilayer perceptron (MLP) networks, our method offers significant computational advantages in models with extensive parameter sharing, such as transformers, convolutional neural networks (CNNs), and recurrent neural networks (RNNs). In these architectures, individual parameters are reused multiple times during both the forward and backward passes. For instance, convolutional filters in CNNs are applied across numerous spatial locations in an image, leading to computational costs proportional to the product of the image's width and height. Similarly, in transformers, attention mechanisms involve operations whose computational complexity scales quadratically with the sequence length, as attention matrices are applied to each pair of elements in the sequence.

In standard backpropagation, the computational cost is influenced not only by the number of parameters but also by the number of times each parameter is applied during training, which is directly related to the input size. This repeated application results in increased computational overhead, particularly in models handling large inputs or long sequences. For example, the computational complexity per parameter in the attention layer of transformers is $O(\text{sequence length}^2)$, and in CNNs, it scales with the number of pixels in the feature maps.

Our Algorithm 1 mitigates this issue by decoupling the computational cost from the input size. Since the Jacobian is precomputed and the parameter updates do not require backpropagation through the network layers, the number of times a parameter is reused does not impact the overall computational complexity. Consequently, in models with significant parameter sharing, our method can substantially reduce training time compared to standard backpropagation.

By eliminating the dependence on input size in the computational cost, our algorithm becomes especially advantageous for training large language models (LLMs), CNNs processing high-resolution images, and RNNs dealing with long sequences. This efficiency gain makes our approach not only theoretically appealing but also practically beneficial for large-scale machine learning tasks where computational resources are a critical consideration.

# 4 RESULTS

## 4.1 LOW RANK USING REFERENCE SET

In this experiment, we investigate the presence of a low-rank structure in the Jacobian by analyzing the behavior of the *rank-to-model size* ratio as the reference set size increases. We compare two networks: a small network with 36K parameters and a larger network with 170K parameters. As discussed in Section 3.2, examining the *rank-to-model size* ratio effectively verifies the low-rank structure. Our results, shown in Figure 2, reveal a clear distinction between the two models. The small network exhibits a steadily increasing *rank-to-model size* ratio, reaching approximately $50\%$, indicating that many parameters contribute to the rank and suggesting the absence of a low-rank structure. In contrast, the larger network maintains a near-zero rank-to-model size ratio, confirming its low-rank behavior as the model size dominates the rank growth. These findings suggest that in models with a low-rank Jacobian, the *rank-to-model size* ratio remains small even as the reference set size increases, allowing for efficient representation of the Jacobian with minimal computational cost. Thus, the *rank-to-model size* ratio serves as a valuable metric for identifying low-rank structures and optimizing network computations.

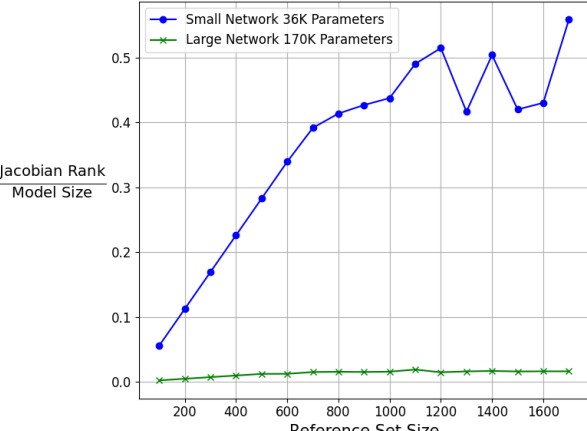

Figure 2: **The $\frac{\text{Jacobian rank}}{\text{model size}}$ ratio as a function of reference set size.** The small network (36K parameters) shows a higher ratio, indicating a lack of low-rank structure, while the large network (170K parameters) maintains a near-zero ratio, confirming its low-rank behavior.

## 4.2 RELATIVE CHANGE IN THE JACOBIAN

In this experiment, we analyzed the relative change in the Jacobian by calculating the ratio between the *Hessian's Operator Norm* and the *Jacobian's Operator Norm*, as we suggested in section 3.2. The goal is to investigate how this relative change behaves as the width of the neural network increases, and its effect on the model's training dynamics.

We conducted the experiment on a 2-layer linear network with increasing widths, ranging from 40K to 400M parameters. Our results, presented in figure 3, illustrate a clear pattern: as the network width increases, the relative change of the Jacobian decreases. Notably, when this relative change falls below $\frac{1}{\sqrt{m}}$, the network enters the "lazy regime," where parameter updates become minimal during training, effectively preventing the network from encountering saddle points. This behavior enabled our algorithm to stabilize and train efficiently, ultimately leading to strong performance, with the model achieving an accuracy of approximately 87% on the MNIST dataset. These findings align with the theoretical results in (Liu et al., 2020), which suggest that networks with larger widths exhibit near-constant tangent kernels due to the small Hessian norm. The graph in figure 3 illustrates the observed trend between 40K and 1M parameters, while the extended experiment covering widths up to 400M parameters is included in the appendix.

Monitoring the relative change in the Jacobian can thus serve as a key indicator of the network's behavior during training. When the relative change is sufficiently small, it signifies that the network is in the lazy regime, allowing for more predictable and controlled training dynamics. By leveraging this observation, we can fine-tune network width and training strategies to optimize performance.

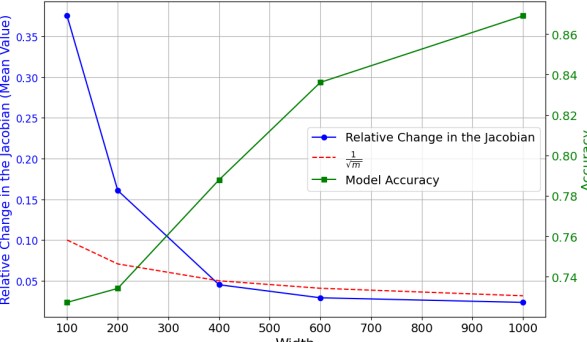

Figure 3: **The relationship between the relative change in the Jacobian and the model width (blue plot) and the corresponding accuracy (green plot) on the MNIST dataset.** As the width of the network increases, the relative change in the Jacobian decreases, indicating the network's entry into the "lazy regime" where training stabilizes and parameters change minimally. This behavior enables efficient training and results in higher accuracy, reaching approximately 87% on the MNIST dataset.

## 5 CONCLUSIONS

We have introduced a novel approach that uniquely combines Neural Tangent Kernel (NTK) theory with the low-rank structure of the Jacobian in large neural networks. This dual focus distinguishes our work as the first to leverage these two aspects concurrently, enabling us to develop a mathematically grounded and computationally efficient backpropagation-free training method operating within the lazy regime. Beyond theoretical advancements, we have provided practical tools for verifying the key assumptions of laziness and low-rank Jacobian structures in specific models and problems, regardless of model width. This comprehensive approach enhances the adaptability of our method, making it applicable to a wide range of real-world applications.

By empirically verifying the laziness and low-rank assumptions, we open a new avenue for alternative training methods that circumvent the need for backpropagation. Our assumption verification methods equip practitioners with practical tools to assess the suitability of our algorithm for their specific models and datasets, ensuring its effectiveness across diverse settings.

The innovative impact of our work lies not only in the theoretical advancement but also in the potential for significant computational efficiency gains. In models with massive parameter-sharing—such as convolutional neural networks and transformer architectures—our algorithm can potentially outperform standard backpropagation by significantly reducing computational overhead and training time.

Future research will focus on extending our approach to these types of models, where parameter-sharing is prevalent. By adapting our algorithm to exploit the structural properties of such networks, we aim to achieve even greater efficiency improvements, making large-scale training more feasible and accessible. Additionally, exploring techniques to enhance the low-rank approximation and optimize the selection of the reference set could further improve performance.

In conclusion, our work provides a promising alternative to traditional backpropagation. By uniting NTK theory with the low-rank Jacobian structure and offering tools for assumption verification, we contribute to both the theoretical understanding and practical advancement of efficient neural network training methods. This opens the door for future exploration and application in models where traditional training methods are computationally prohibitive.

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

# A APPENDIX

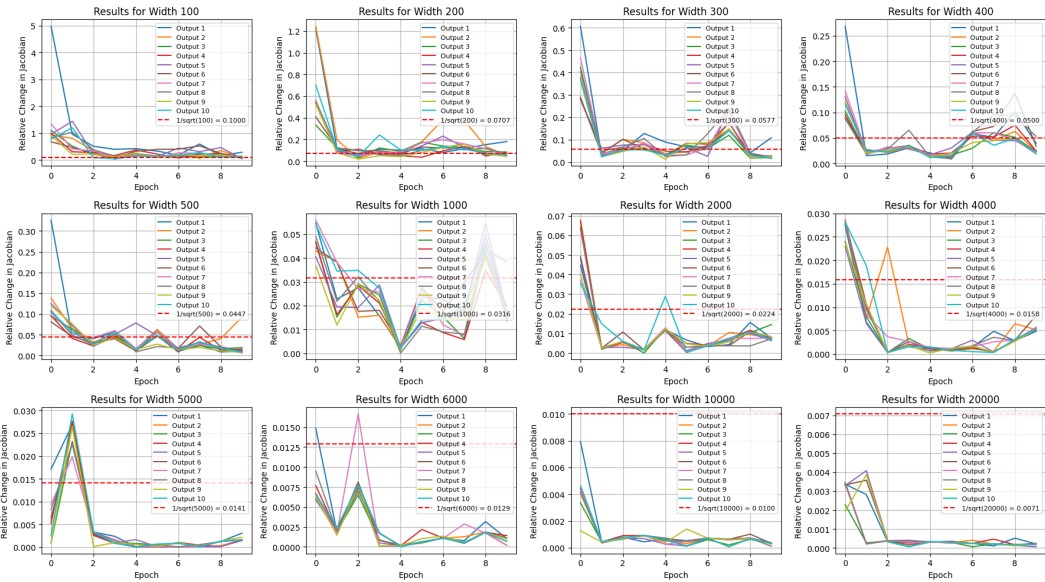

Figure 4: Extended Results for Relative Change in the Jacobian for Varying Neural Network Widths. The graphs show the relative change in the Jacobian over the first 10 epochs for a wider range of network widths, from 100 to 20,000, as a continuation of the main analysis presented in section 3.2. The relative change is calculated as the ratio between the *Hessian's Operator Norm* and the *Jacobian's Operator Norm* for each output. The dashed red line in each plot indicates the threshold $\frac{1}{\sqrt{m}}$ (where $m$ is the width), below which the network enters the "lazy regime." As observed, increasing the network width leads to a decrease in the relative change of the Jacobian. This extended analysis, including widths up to 400M parameters, further reinforces the findings that wider networks tend to enter the lazy regime more quickly.

