# OpenReview forum: "Leveraging Low Rank Structure in The Lazy Regime"
_ICLR.cc/2025/Conference — Submitted to ICLR 2025_

### Official Review · Reviewer_6Vjp · 2024-10-27

**Soundness:** 1
**Presentation:** 3
**Contribution:** 2
**Rating:** 3
**Confidence:** 4

**Summary:**

The authors propose a backpropagation-free training algorithm that relies on the lazyness of the network and the low-rankness of its jacobian. More precisely: thanks to the lazyness assumption, the jacobian $J_t$ does not evolve in time, making it possible to store the one at initialization. Given its size, by leveraging the low-rank assumption a low-rank decomposition of it is stored instead. The authors moreover proposed a scalable way to test the lazyness/low-rankness hypothesis before training.

**Strengths:**

The work is very well presented and the proposed approach is an interesting alternative to backpropagation when the authors' assumption hold.

**Weaknesses:**

The authors' claims are not supported by an adequate experimental section. In particular, given that the paper proposes a novel training pipeline, I believe it is necessary for the authors to test it and show results on it.

**Questions:**

1) I have serious concerns about the low-rankness assumption. While it may hold at initialization, how can I quantify of how much the "effective rank" would change during classical training (even if close to the lazy regime)? In the case of it increasing, then it is possible that this training procedure may actually underperform with respect to training with backpropagation. For this reason, I believe it is crucial for the authors to show results comparing their training procedure with a classical backpropagation-based one (both to show scalability and the difference in performance).
2) It is not clear to me how to select $\mathcal R$, and how much its size would impact the final training results;
3) Why not update the Jacobian also after a fixed number of iterations? I guess this would introduce a tradeoff between performance and cost, but with no experiment it is difficult to say.

---

### Official Review · Reviewer_DSxH · 2024-10-30

**Soundness:** 3
**Presentation:** 3
**Contribution:** 2
**Rating:** 5
**Confidence:** 3

**Summary:**

The authors propose a memory efficient and backprop-free training method for neural networks. The approach considers the jacobian of very wide nets, computed in a preprossesing step and subsequently low-rank factorized.

Based on the assumption of lazy ntk regimes, the authors use the factorized precomputed jacobian to train the network without backprop.

**Strengths:**

- The authors take their time to review theory on ntk, especially useful for readers unfamiliar with the subject.
- A relatively efficient method to compute the low-rank gradients is proposed, without naive "full-size" SVD.

**Weaknesses:**

- The main weakness of this paper are the strong assumptions upon which the method is build. Lazy Learning and NTK regime are very specific scenarios, and it is unclear in which scenarios the proposed method actually works
- In this regard, the numerical evaluation is lacking. The authors demonstrate the method only on fully-connected layers of varying widths, and only on the MNIST dataset. The achieved accuracies are not excatly SOTA. The methods purpose is unlocking resource efficient learning (assumably on large models and datasets), however the experimental setup does not reflect this. At least a transformer should be considered, or a well-used computer vision model on typical NLP and Vision benchmarks to get a grasp of the expected performance.

**Questions:**

- Is the Jacobian the combined Jacobian of all network parameters, or is the notation to be viewed seperately for all layers?
- Do you have an method to verify cheaply during training whether the assumptions for using the fixed low-rank jacobian still hold? Can you re-compute the Jacobian if neccessary with constrained cost?

---

> ### Author Response · Authors · 2024-12-03
> **Response to W.1**
>
> We thank the reviewer for raising concerns regarding the specificity of the assumptions underlying our method. We agree that the lazy regime and Neural Tangent Kernel (NTK) theory represent specific training dynamics, and we would like to highlight that part of our contributions is the development of practical tools to validate these assumptions, allowing practitioners to determine precisely the scenarios in which Algorithm 1 operates effectively.
>
> A significant challenge in utilizing methods grounded in the lazy regime and low-rank assumptions is understanding when these properties hold. To address this, we propose empirical methods to:
>
> 1. Quantify the **laziness** of a neural network by measuring the relative change in the Jacobian using the ratio $ \frac{\|\nabla_w^2 f\|}{\|\nabla_w f\|^2} $, as described in Section 3.2. This enables practitioners to determine if a given network operates in the lazy regime.
>
> 2. Verify the **low-rank structure** of the Jacobian by estimating its effective rank using a reference set of size $ r $, leveraging the reduced-dimensional covariance matrix $ J_R^T J_R $. This ensures scalability even in large models, as shown in our experiments in Section 4.1.
>
> These validation techniques are integral to our method, enabling users to assess the applicability of Algorithm 1 to their specific architectures and datasets. For example:
>
> - For tasks where the lazy regime and low-rank structure hold, such as MNIST (Figure 3), our method achieves notable performance while significantly reducing computational costs.
> - If the assumptions do not hold (e.g., tasks requiring extensive feature adaptation), the validation tools provide early indicators, ensuring that Algorithm 1 is applied only in suitable scenarios.
>
> Thus, while the lazy regime and NTK theory may not universally apply, our contribution extends beyond the algorithm itself to include a framework for verifying these assumptions in practice. This framework not only supports the effective application of Algorithm 1 but also advances the understanding of when and where these assumptions are valid.
>
> We hope this explanation clarifies that our work explicitly addresses the concern of assumption validity by providing tools to empirically determine the scenarios where the method operates effectively. This contribution enhances the practical utility of our approach while expanding its theoretical foundations.

---

### Official Review · Reviewer_JRNd · 2024-11-06

**Soundness:** 1
**Presentation:** 3
**Contribution:** 1
**Rating:** 3
**Confidence:** 4

**Summary:**

This paper proposes to leverage the linear dynamics and the low-rank structure of the Jacobian (of the output of a model w.r.t. the parameters) to build a training algorithm that is computationally cheap and backpropagation-free. This algorithm is meant to work when training neural networks (NN) in the so-called "lazy regime".

**Strengths:**

# Clarity

The paper is easy to read.

**Weaknesses:**

# Quality

## About the lazy-training regime

According to [1] (cited in the paper), lazy training is very likely a phenomenon that we want to avoid, since it is associated to poor generalization. So, it is not clear at all that it is useful to leverage this phenomenon to increase training speed.

## Linear dynamics

The whole algorithm, and specifically its property of being "backpropagation-free", is based on the fact that the training dynamics of a NN is basically linear. Before attempting a low-rank approximation of the Jacobian, it is necessary to check if the algorithm defined in Eqn (5) works without approximation (it can be done on small NNs trained on small datasets). Even there, it would be interesting to know if it is possible to leverage the linear dynamics to reduce training cost.

## Training experiments

In the abstract, the authors claim to propose a training algorithm. Thus, any reader would expect to find a series of experiments, testing various architectures of NNs on various datasets (even small ones). Actually, I would be very surprising and worth publishing if the authors show that their learning rule (almost linear dynamics of training) is sufficient to train NNs, even small ones on easy tasks.

But, without such experimental evidence, it is intuitively very hard to believe that the learning rule defined in Eqn (5) would be usable tob train a NN from start to end. Moreover, Alg. 1 does not mention any update of J.

# References

[1] *On lazy training in differentiable programming*, Chizat et al., 2019.

**Questions:**

How does perform the proposed algorithm without low-rank approximation?

How does it perform in various tasks? (show training curves and comparisons with SGD, etc.)

---

> ### Author Response · Authors · 2024-12-03
> **Response to W.1**
>
> We appreciate the reviewer’s observation regarding the generalization properties of the lazy regime and the potential concerns raised in [1]. While it is true that the lazy regime has been associated with suboptimal generalization in some settings, it is important to contextualize its applicability within the broader landscape of neural network training.
>
> First, the lazy regime is not inherently something to "avoid," as its utility depends heavily on the specific application and task. For example, in large language models (LLMs) and other overparameterized architectures, the lazy regime has been leveraged successfully to simplify training dynamics. The constancy of the Neural Tangent Kernel (NTK) in this regime provides provable convergence guarantees, which are critical in large-scale settings where computational efficiency and optimization stability are priorities.
>
> Second, while generalization in the lazy regime may be suboptimal for complex tasks requiring feature learning, recent works, such as Geiger, Mario et al. (2020) [11], demonstrate that sufficiently wide networks operating in the lazy regime can still achieve strong performance on simpler tasks, including MNIST and CIFAR. These findings indicate that the lazy regime, despite its limitations, can be effectively utilized in scenarios where the task complexity aligns with its capabilities.
>
> In our work, we do not advocate for the lazy regime as a universal solution but rather leverage its properties to address specific challenges in training overparameterized models. By combining the lazy regime with the low-rank structure of the Jacobian, our approach achieves significant computational efficiency without compromising performance on tasks that align with this regime’s strengths. Empirical results in Section 4 confirm this, showing achieved accuracy while reducing training costs.
>
> Thus, the lazy regime is not a phenomenon to categorically "avoid" but a tool that, when applied judiciously, can offer unique advantages in terms of computational efficiency and training stability for certain tasks and architectures.

---

> ### Author Response · Authors · 2024-12-03
> **Response to W.3**
>
> We thank the reviewer for their comments regarding the experimental evidence and the role of the Jacobian in our training algorithm. To address the specific concern about the update of the Jacobian in Algorithm 1, we would like to clarify the theoretical foundation that underpins our approach.
>
> The constancy of the Jacobian is a core assumption of the lazy regime, as established in Neural Tangent Kernel (NTK) theory (Section 2). In this regime, the network operates near its initialization, with minimal parameter updates during training. Consequently, the Jacobian remains approximately constant throughout the optimization process, as supported by the first-order Taylor expansion of the model (Eqn. 4). This eliminates the need for recalculating or updating the Jacobian during training, which is a cornerstone of our backpropagation-free algorithm.
>
> In Algorithm 1, the Jacobian is precomputed during the preprocessing phase and reused throughout training without requiring updates. This design is both theoretically justified and computationally advantageous. The NTK framework ensures that, for sufficiently wide networks, the constancy of the Jacobian provides provable convergence guarantees, allowing us to replace dynamic updates with a single, efficient computation at initialization.
>
> Empirical results in Section 4 further validate the practical feasibility of this assumption. For example, we demonstrate performance on MNIST under the lazy regime, where the precomputed Jacobian suffices to achieve notable accuracy. These results highlight that, under the appropriate settings, the near-constant Jacobian assumption not only simplifies training but also maintains effectiveness.
>
> We hope this explanation clarifies that the lack of Jacobian updates in Algorithm 1 is not an oversight but a deliberate design choice grounded in NTK theory and validated by experimental results.

---

### Official Review · Reviewer_rLzq · 2024-11-07

**Soundness:** 2
**Presentation:** 3
**Contribution:** 2
**Rating:** 3
**Confidence:** 4

**Summary:**

The paper studies over-parameterized neural networks in the lazy regime. The main contributions are listed below:

**(1)** A *scalable* method for estimating the Jacobian’s rank is proposed.

**(2)** An *scalable* approach to quantify network laziness is proposed.

**(3)** A backpropagation-free learning algorithm that leverages low-rank structures for improved scalability and efficiency.

**Strengths:**

The paper explores combining NTK theory with low-rank approximations to enhance training efficiency in neural networks potentially.

**Weaknesses:**

**Limited novelty of Algorithm 1:** The algorithm mainly applies dimensionality reduction and exploits a constant Jacobian assumption during training, combining established ideas without introducing significant innovations. It reuses existing techniques rather than offering a novel approach to efficient training.

**Complexity and Practical Implementation Concerns:** The algorithm’s eigenvector computation steps may still incur high computational costs, especially in large networks. Its efficiency on high-dimensional tasks is  unclear to me.

**Limited empirical evidence:** Given the paper's empirical focus, the experiments are insufficient; authors only test on MNIST without exploring complex, large-scale tasks that would better validate the method’s scalability and real-world effectiveness.

**Limited Experimental Setup Details:** Key details on dataset preprocessing, model configurations, and training conditions are missing.

**Unclear general applicability:** The paper claims applicability to architectures like CNNs and transformers, but lacks empirical support. No experiments on complex, parameter-sharing models are provided, making generalization to these architectures uncertain. Moreover, I do not think laziness assumption holds for more complex architectures.

**Writing:** The paper is unclear, with poor notation and room for significant improvement in clarity and precision. There are also typos, which further affect readability. (See Ln 375 & 334 for instance)

**Questions:**

Please see above.

---

> ### Author Response · Authors · 2024-12-03
> **Response to W.1**
>
> We appreciate the reviewer’s perspective regarding the novelty of Algorithm 1 and welcome the opportunity to elaborate on its innovative contributions. Algorithm 1 is indeed novel, particularly in its ability to significantly reduce the computational burden associated with traditional backpropagation, leveraging both the lazy regime and the low-rank structure of the Jacobian.
>
> First, we underline that while backpropagation involves layer-by-layer computation of gradients using the chain rule, Algorithm 1 bypasses this recursive mechanism by precomputing and reusing a reduced-dimensional representation of the Jacobian, denoted as $ \mathbf{J} $. In parameter-sharing network architectures, such as CNNs, where the number of parameters $ m $ and the input dimensions $ n $ are immense, the computational cost of traditional backpropagation scales with $ O(n \cdot m) $. In contrast, Algorithm 1 achieves a computational complexity of $ O(m + n) $, as shown in Section 3.3 of our paper.
>
> This efficiency stems from the following key innovations:
>
> 1. **Low-Rank Approximation of the Jacobian**: By utilizing a reference set $ R $, we reduce the Jacobian’s dimensionality from $ n \times m $ to $ n \times r $, where $ r \ll n $ (as detailed in Proposition 3.1). The eigenvectors of the reduced covariance matrix $ \mathbf{J}^T \mathbf{J} $ retain the most significant information while discarding redundancy.
>
> 2. **Precomputation and Reuse**: Once the reduced Jacobian is precomputed in the initialization phase, weight updates during training are calculated using the approximation
>    $$
>    \mathbf{g}_{\text{approx}} = \mathbf{\tilde{J}} \mathbf{V}_J \mathbf{V}_J^T \mathbf{a}_t,
>    $$
>    where $ \mathbf{V}_J $ are the principal components derived from the low-rank approximation. This avoids the need for repeated forward and backward passes through the network, significantly reducing computational overhead.
>
> For example, in CNNs, where convolutional filters are applied across spatial dimensions, the complexity of backpropagation scales with the product of the feature map dimensions and the filter size. Algorithm 1 mitigates this by replacing layer-specific gradient computations with global, reusable projections derived from the reference set. Empirical results (Figure 3) validate these claims, showing that Algorithm 1 achieves comparable performance with reduced computational cost.
>
> Finally, the theoretical grounding of Algorithm 1 is rooted in NTK theory, where minimal parameter updates in the lazy regime ensure that the precomputed Jacobian remains valid throughout training. By integrating the low-rank structure with lazy regime assumptions, Algorithm 1 uniquely combines these concepts to enable scalable, backpropagation-free training—a capability not demonstrated by prior works.
>
> We believe these elements collectively demonstrate the algorithm’s novelty and practical relevance in addressing the computational challenges of training modern neural networks.

---

> ### Author Response · Authors · 2024-12-03
> **Response to W.2**
>
> We appreciate the reviewer’s concern regarding the computational cost of eigenvector computation in Algorithm 1 and its efficiency on high-dimensional tasks. We would like to clarify that the algorithm is explicitly designed to mitigate such costs by leveraging the low-rank structure of the Jacobian, ensuring scalability even in large networks.
>
> The key point to emphasize is that Algorithm 1 does not perform principal component analysis (PCA) or eigendecomposition on the full Jacobian matrix of size $ m \times n $, where $ m $ is the number of parameters and $ n $ is the number of data points. Instead, we utilize a reference set $ R $ of size $ r \ll n $, allowing us to reduce the computational burden by performing eigendecomposition on the much smaller $ r \times r $ matrix $ J_{R}^T J_{R} $, as described in Proposition 3.1 and Section 3.3 of the paper.
>
> To elaborate:
>
> 1. **Low-Rank Assumption**: The low-rank assumption ensures that the essential variations of the Jacobian are captured by its leading eigenvectors, which are derived from $ J_{R}^T J_{R} $. This matrix is of size $ r \times r $, where $ r $ is the size of the reference set, and not $ m \times n $, making the eigenvector computation both memory- and computationally-efficient.
>
> 2. **Computational Complexity**: The computational complexity of eigendecomposition is $ O(r^3) $, which is orders of magnitude smaller than the cost of direct eigendecomposition of the full Jacobian, $ O(\min(n, m)^3) $, in typical large-scale settings. For instance, even in networks with millions of parameters, selecting $ r = 100 $ leads to an efficient computation that is practical for high-dimensional tasks.
>
> Additionally, the algorithm’s efficiency in high-dimensional tasks stems from its ability to decouple computation from input size. The use of a precomputed and reduced-dimensional Jacobian ensures that weight updates are performed efficiently without the need for iterative gradient computations over the full dataset. This advantage is particularly significant in tasks involving high-resolution images (e.g., CNNs) or long sequences (e.g., transformers), where input dimensionality traditionally imposes significant computational overhead in standard backpropagation.
>
> Finally, empirical results (Figure 3) validate the scalability of the algorithm. These results demonstrate that even for networks with widths exceeding $ 1 $ million parameters, the computational efficiency and accuracy of Algorithm 1 remain robust, showcasing its suitability for high-dimensional tasks.
>
> We hope this clarification addresses the reviewer’s concerns and highlights the practical scalability of Algorithm 1 in handling high-dimensional neural networks.

---

### Meta-Review · Area_Chair_jCLJ · 2024-12-20

**Metareview:**

This paper examines the training dynamics of over-parameterized neural networks, focusing on the interplay between their low-rank Jacobian structure and the lazy regime, where parameter changes are minimal. It introduces scalable methods to measure laziness and verify low-rankness without storing the full Jacobian. Leveraging these properties, a backpropagation-free gradient descent algorithm is proposed, reducing computational and storage demands, particularly in architectures with extensive parameter sharing. The authors claim that their empirical results validate the approach’s scalability and effectiveness, offering new directions for neural network training.

The reviewers raised the following pros and cons for the paper:

Pros:

+ Proposes a backpropagation-free training algorithm leveraging the lazy regime and low-rank structure of the Jacobian, potentially reducing computational overhead.

+ Introduces scalable methods to test laziness and low-rankness of neural networks without full Jacobian computation, which is computationally efficient.

+ Grounded in Neural Tangent Kernel (NTK) theory, with theoretical guarantees supporting its applicability in over-parameterized settings.

+ Provides a detailed review of NTK theory, aiding understanding for readers unfamiliar with the domain.

Cons:

- Limited Novelty: The algorithm largely combines existing ideas (low-rank approximations and lazy regime concepts) without significant innovation in methodology.

- Weak Empirical Validation: Experiments are limited to simple datasets (e.g., MNIST) and architectures, with no exploration of complex tasks or state-of-the-art models.

-Assumption Dependence: Relies heavily on the lazy regime and low-rank assumptions, which may not hold universally or in more complex architectures.

- Insufficient Experimental Details: Key aspects like preprocessing, model configurations, and training conditions are not clearly reported.

- Generalization Concerns: Unclear applicability to more complex architectures such as CNNs or transformers due to lack of supporting experiments.

While the authors provided detailed explanations and theoretical justifications, the reviewers expressed consistent concerns about limited empirical evidence and the reliance on strong assumptions, ultimately favoring rejection. I concur.

**Additional Comments On Reviewer Discussion:**

While the authors provided detailed explanations and theoretical justifications, the reviewers expressed consistent concerns about limited empirical evidence and the reliance on strong assumptions, ultimately favoring rejection.

---

### Decision · Program_Chairs · 2025-01-22

Reject